# The Oxysterol 25-Hydroxycholesterol Inhibits Replication of Murine Norovirus

**DOI:** 10.3390/v11020097

**Published:** 2019-01-24

**Authors:** Ghada T. Shawli, Oluwapelumi O. Adeyemi, Nicola J. Stonehouse, Morgan R. Herod

**Affiliations:** School of Molecular and Cellular Biology, Faculty of Biological Sciences and Astbury Centre for Structural Molecular Biology, University of Leeds, Leeds LS2 9JT, UK; G.Shawli@leeds.ac.uk (G.T.S.); O.O.Adeyemi@leeds.ac.uk (O.O.A.); N.J.Stonehouse@leeds.ac.uk (N.J.S.)

**Keywords:** 25-HC, Murine norovirus, Nystatin, MNV, Replication, Apoptosis

## Abstract

Cholesterol, an essential component of mammalian cells, is also an important factor in the replicative-cycles of several human and animal viruses. The oxysterol, 25-hydroxycholesterol, is produced from cholesterol by the enzyme, cholesterol 25-hydroxylase. 25-hydroxycholesterol (25-HC) has been shown to have anti-viral activities against a wide range of viruses, including a range of positive-sense RNA viruses. In this study, we have investigated the role of 25-HC in norovirus replication using murine norovirus (MNV) as a model system. As a control, we employed herpes simplex virus-1 (HSV-1), a pathogen previously shown to be inhibited by 25-HC. Consistent with previous studies, 25-HC inhibited HSV-1 replication in the MNV-susceptible cell line, RAW264.7. Treating RAW264.7 cells with sub-cytotoxic concentrations of 25-HC reduced the MNV titers. However, other sterols such as cholesterol or the oxysterol, 22-S-hydroxycholesterol (22-S-HC), did not inhibit MNV replication. Moreover, treating MNV-infected RAW264.7 cells with 25-HC-stimulated caspase 3/7 activity, which leads to enhanced apoptosis and increased cell death. Our study adds noroviruses to the list of viruses inhibited by 25-HC and begins to offer insights into the mechanism behind this inhibition.

## 1. Introduction

Human norovirus (HNV) is the leading cause of non-bacterial gastroenteritis worldwide in people of all ages [1]. HNV is transmitted through the faecal–oral route, primarily via interaction with infected individuals and contact with contaminated food or water [2,3]. In high income countries, HNV infection causes a substantial economic burden and drain on healthcare services, whereas in low and lower-middle income counties, HNV infections result in significant mortality. Despite the medical importance of these viruses, little is known about the mechanism of pathogenesis and there are no approved therapies or vaccines to treat infection.

Noroviruses are part of the *Caliciviridae* family of positive-sense RNA viruses [4,5]. The viral genome is translated into at least three open reading frames (ORFs) that encode the viral non-structural polyprotein and the two viral structural proteins, VP1 and VP2 [4]. An understanding of the replication and pathogenesis of HNV has been hindered, in part, due to the difficulty in culturing these viruses in the laboratory [6,7]. Thus, closely related animal caliciviruses, such as feline calicivirus (FCV) [8] and murine norovirus (MNV) [9], have been valuable models for studying the basic molecular biology of this family of viruses.

Cholesterol and related sterols are vital lipid components of eukaryotes that have been shown to play important roles in the replicative-cycles of multiple human and animal viruses. Oxysterols, the oxidised derivatives of cholesterol, play important roles in a variety of physiological processes including sterol transportation, the regulation of cholesterol homeostasis and innate immunity. They are also involved in the progression of a wide range of diseases and have emerged as compounds that antagonise the replication of numerous viruses. The oxysterol, 25-hydroxycholestrol (25-HC), is synthesised from cholesterol by the enzyme, cholesterol-25-hydroxylase (CH25H), which is encoded by the interferon-stimulated gene (ISG) *ch25h* [10]. The enzyme, CH25H, and its product, 25-HC, have been demonstrated to possess anti-viral activities against a wide range of viruses, both enveloped [11,12,13,14,15,16,17,18,19] and non-enveloped [20,21,22]. For example, among enveloped viruses, studies have shown that 25-HC can inhibit viral attachment [11] and entry into the cells [11,12,16,22,23], transcription and protein synthesis [11], viral genome replication [12,13,15], membranous replication factory formation [24] and virion production [14]. 25-HC can also inhibit the post-entry step of a number of viruses such as hepatitis C virus, by blocking the activation of sterol regulatory element-binding protein (SREBP) [25], a transcription factor required for lipid and cholesterol biosynthesis. For non-enveloped viruses, 25-HC is thought to interact together with oxysterol-binding protein, resulting in reduced cholesterol accumulation in the membranous scaffolds of viral replication complexes and thus inhibit virus replication and entry into cells [22,26,27]. The diversity of the viruses inhibited by 25-HC makes this oxysterol an attractive starting point for the development of future pan-viral therapeutic approaches. 

In this study, we have conducted the first investigation of the effect of 25-HC on noroviruses, using the MNV model system. Our data suggest that 25-HC has an inhibitory effect on MNV replication, potentially at multiple stages of the replicative-cycle and can stimulate an MNV-induced apoptotic response. 

## 2. Materials and Methods

### 2.1. Reagents

The cholesterol and oxysterols were reconstituted in 5.5 mg/mL ethanol (13.5 mM) for 25-HC and 22-S-HC, and 5.2 mg/mL ethanol (13.5 mM) for cholesterol (all Sigma-Aldrich). Nystatin (Sigma-Aldrich) was prepared in 50 mg/mL dimethyl sulphoxide (DMSO) (54 mM). All compounds were stored at −20 °C.

### 2.2. Cell Lines and Viruses

Mouse leukemic macrophage RAW264.7 cells (gifted by Ian Clarke, University of Southampton, UK) were maintained in high-glucose Dulbecco’s modified Eagle’s medium (DMEM) supplemented with 10% *v*/*v* foetal calf serum (FCS), 50 U/mL penicillin (Sigma-Aldrich), 50 μg/mL streptomycin (Sigma-Aldrich) and 24 mM HEPES buffer (Sigma-Aldrich) at 37 °C in 5% CO_2_.

GFP-labelled herpes simplex virus-1 (HSV-1–GFP) was kindly provided by Chris Jones (University of Leeds, UK). MNV-1 strain CW1P3 [28] used in this study was recovered from an infectious clone. The MNV stocks were propagated in RAW264.7 cells and incubated for 48–72 h at 37 °C in 5% CO_2_. When the full cytopathic effect (CPE) was observed, the cells and supernatants were harvested, and the virus was released through three freeze (−80 °C)/thaw (25 °C) cycles. The supernatant was clarified by centrifugation to remove cellular debris and stored at −80 °C. The titer of the viral stocks was determined using the median tissue culture infective dose (TCID_50_) assay as previously described [29]. The viral titers were calculated using the Spearman and Kärber algorithm and expressed as TCID_50_/mL [30,31].

To assess the effect of 25-HC on the viral capsid, 100 µL non-purified stocks of MNV were incubated in the presence of 25-HC (up to a final concentration of 135 µM) for 4 h at 37 °C. As a solvent-only control, the virus was also incubated with ethanol-alone (final concentration 1% *v*/*v*). To control for any 25-HC-related toxicity, growth media without the virus were also incubated with the same range of 25-HC concentrations. After incubation, the virus and media were diluted in DMEM to a final volume of 1000 µL. The titer of the diluted virus was assessed by the TCID_50_ assay on RAW264.7 cells.

### 2.3. Assessment of Cell Viability

Cells seeded in 96-well plates (5 × 10^4^ cells/well) were incubated at 37 °C in 5% CO_2_ for 16 h. A serial dilution of each compound was added to six replicate wells per dilution and untreated control cells were treated with solvent-only (ethanol or DMSO, as appropriate, at the final concentration of 0.1% *v*/*v*). The cells were treated for 72 h before assessing cell viability by MTS (3-(4,5-dimethylthiazol-2-yl)-5-(3-carboxymethoxyphenyl)-2-(4-sulfophenyl)-2H-tetrazolium) assay using the Cell Titer 96 AQueous One Solution (Promega), according to the manufacturer’s instructions. This assay measures cell viability by quantifying the NADPH/NADH activity that is produced by dehydrogenase enzymes in the metabolically active cells, which reduces the MTS compound into a colored formazan product. Briefly, to each well 10 µL Cell Titer 96 AQueous One reagent was added, the plates were then incubated at 37 °C for 2 h and the absorbance was read at 490 nm using a microplate reader. The absorbance background was corrected by subtracting the absorbance from the wells containing medium only. The results are expressed as the percentage viability normalized to the solvent-only control cells (100% viability) and wells containing the medium alone (0% viability). Nonlinear dose–response curves were fitted to the mean averaged data points using GraphPad Prism 7 software without constraints. 

### 2.4. MNV Dose–Response Curves

RAW264.7 cells were seeded in 12-well plates (4 × 10^5^ cells/well) and incubated at 37 °C in 5% CO_2_ for 16 h. Replicate wells were treated with a serial dilution of 25-HC (up to a final concentration of 135 µM), nystatin (up to a final concentration of 500 µM) or ethanol or DMSO solvent-only control (final concentration 0.1% *v*/*v*) and immediately infected with MNV at a multiplicity of infection (MOI) of 0.05 TCID_50_/cell. As controls, the wells were infected with MNV without the addition of the compounds and treated with the various compounds without the addition of the virus. After 48 h, the virus was harvested by three freeze (−80 °C) and thaw (25 °C) cycles. The samples were clarified by centrifugation and the virus was titrated by TCID_50_ assay on RAW264.7 cells. Nonlinear dose–response curves were fitted to the mean averaged data points using GraphPad Prism 7 software without constraints.

### 2.5. Single Cycle Growth Analysis

RAW264.7 cells seeded in 12-well plates (4 × 10^5^ cells/well) were incubated at 37 °C in 5% CO_2_ for 16 h. Replicate wells were treated with 25-HC (6.6 µM), nystatin (50 µM) or the ethanol-only control (final concentration of 0.1% *v*/*v*) and immediately infected with MNV at a MOI of either 0.05 or 5 TCID_50_/cell, as indicated in the figure legend. The infected untreated cells, mock cells and cells treated with only 25-HC were included as controls. The virus was harvested from replicate wells at regular intervals for up to 24 h post-infection. The samples were clarified by centrifugation following three freeze (−80 °C) and thaw (25 °C) cycles and the virus was titrated by TCID_50_ assay on RAW264.7 cells.

### 2.6. Fluorescent Focus Assay

RAW264.7 cells seeded in 96-well plates (5 × 10^4^ cells/well) were infected with a 10-fold serial dilution of MNV stocks. For each experiment, four replicate wells were infected in the presence of 25-HC (6.6 µM), nystatin (50 µM) or ethanol as a solvent-only control (final concentration 0.1% *v*/*v*). As additional controls, the replicate wells were treated with 25-HC, nystatin or ethanol-alone and left uninfected. At 12 h post-infection, the cells were fixed with 4% paraformaldehyde, washed with PBS and permeabilized with saponin buffer (0.1% saponin, 10% FCS, 0.1% sodium azide) for 1 h at 4 °C. MNV infection was detected with an anti-non-structural protein 1/2 (NS1/2) antibody (gifted by Ian Clarke, University of Southampton) as previously described [32]. The primary antibody was detected with an anti-mouse Alexa Flour568-conjugated secondary antibody (Thermo Fisher). Images were obtained using IncuCyte ZOOM Dual Colour FLR (Essen Biosciences) and used to measure NS1/2-positive cells using the integrated IncuCyte ZOOM software (Essen Biosciences).

### 2.7. Time-of-Addition Assays

Replicate plates of RAW264.7 cells grown at 37 °C in 5% CO_2_ were pre-treated with 25-HC (6.6 µM), nystatin (50 µM) or the ethanol-only control (0.1% *v*/*v*) for 24 h, 16 h or 4 h prior to infection with MNV at a MOI of either 0.05 or 5 TCID_50_/cell, as indicated in the figure legend. Subsequently, the RAW264.7 cells were infected with MNV and immediately treated with the compounds, or at 4 h post-infection, after removal of the virus. At 48 h post-infection, the virus was harvested, frozen (−80 °C) and thawed (25 °C) three times, clarified by centrifugation and titrated by TCID_50_ assay on RAW264.7 cells.

### 2.8. Measuring Caspase 3/7 Activity

RAW264.7 cells were seeded in 96-well plates (5 × 10^4^ cells/well) and allowed to adhere for 16 h. For each experiment, four replicate wells were infected with MNV in the presence or absence of 25-HC (6.6 µM). Cells treated with 25-HC alone or mock untreated cells were included as controls. A Cell Event Caspase 3/7 Green Detection Reagent (Thermo Fisher) was used to measure caspase 3/7 activity up to 24 h post infection following the manufacturer’s instructions. Images were obtained using IncuCyte ZOOM Dual Colour FLR (Essen Biosciences) and used to measure caspase 3/7 activity by quantifying green fluorescence using the integrated IncuCyte ZOOM software (Essen Biosciences).

### 2.9. Measuring HSV-1–GFP Replication

RAW264.7 cells were seeded in 96-well plates (5 × 10^4^ cells/well) and incubated at 37 °C in 5% CO_2_ for 16 h. The compounds were diluted in a serum-free medium to achieve final concentrations of 50 µM nystatin and 6.6 µM 25-HC, 22-S-HC and cholesterol. To six replicate wells, 100 µL of each compound and HSV-1–GFP was added and the plates were incubated at 37 °C in 5% CO_2_. Six replicate wells of infected untreated cells and mock cells were included as controls. After 48 h, HSV-1–GFP replication was measured by the GFP expression using an IncuCyte Dual Colour ZOOM FLR (Essen Bioscience); with the number of GFP expressing cells enumerated using an internal algorithm in the integrated IncuCyte ZOOM software (Essen Bioscience). The results were normalized to the percentage of HSV-1–GFP infected, untreated cells.

### 2.10. Cytotoxicity of MNV in the Presence of Oxysterols

RAW264.7 cells were seeded in 96-well plates (5 × 10^4^ cells/well) and incubated at 37 °C in 5% CO_2_ for 16 h. Serial dilutions of the MNV stocks (10^0^–10^−6^) were prepared in DMEM supplemented with 50 U/mL penicillin, 50 μg/mL streptomycin and 24 mM HEPES buffer. To groups of six replicate wells, 100 µL of each virus dilution was added. To each group, 25-HC, 22-S-HC, cholesterol (all at final concentrations of 6.6 µM) or nystatin (at a final concentration of 50 µM) was added immediately after the addition of the virus. The final volume of each well was 200 µL. As controls, the wells were infected with MNV without the addition of compounds and treated with the various compounds without the addition of the virus. After 72 h, cell viability was assessed using Cell Titer 96 AQueous One reagent, as described above. 

### 2.11. Statistical Analysis

This was performed using two-tailed unpaired student *t* test or two-way ANOVA with the Bonferroni test using GraphPad Prism 7. Significant differences between treated and untreated cells were expressed with *p*-values < 0.05 (*), <0.01 (**) and <0.001 (***). Error bars in the graphs represent mean ± SEM (standard error of mean) of a number of experiments.

## 3. Results

### 3.1. 25-HC Reduced MNV Titer in a Single-Cycle Growth Curve

The oxysterol, 25-HC, has been demonstrated to have anti-viral effects against a wide range of viruses by targeting several stages of the viral replication cycle [11,12,13,14,16,20,21]. These studies typically use 25-HC in the nanomolar to micromolar range, with different viruses and target cells demonstrating differing sensitivities. Thus, we began by investigating the protective effect of 25-HC on MNV replication in a dose–response experiment. Subsequently, MNV replication was assessed in the presence of nystatin, a well-documented inhibitor of MNV that acts by sequestering cholesterol from the plasma membrane [33]. The compounds remained on the cells for the duration of the experiment and the inhibition of viral replication was determined by TCID_50_ assay on RAW264.7 cells (Figure 1A).

There was a dose-dependent reduction in MNV replication upon the addition of 25-HC or the control inhibitor nystatin (a well-documented inhibitor of MNV endocytosis [33]) with IC_50_ values calculated at of ~2.8 µM and 53 µM, respectively. Complete inhibition of virion production was only observed with the highest concentration of 25-HC or nystatin tested, where cellular cytotoxicity is likely. Therefore, to determine any cytotoxic effect of the compounds used in these assays, MTS viability assays were performed on RAW264.7 cells with the same range of compound concentrations, using the Cell Titer 96 AQueous One reagent (Figure 1B). Both 25-HC and nystatin demonstrated some dose-dependent cytotoxicity, as seen previously [11], with CC_50_ concentrations calculated at approximately 20 µM and 125 µM, respectively. The maximum concentrations tolerated before intolerable cytotoxicity were observed were 6.6 µM (calculated at ~83% viability) and 50 µM (calculated at ~92% viability), respectively. These concentrations were therefore chosen for all subsequent experiments.

Having established the maximum concentration of 25-HC tolerated in our assays we chose to assess the effect of adding exogenous 25-HC on the replication of MNV in a single-cycle growth assay. RAW264.7 cells were infected with MNV at either 0.05 or 5 TCID_50_/cell, before the immediate addition of exogenous 25-HC or ethanol as a solvent-only control. As a positive control, infected cells were treated with nystatin as before. The compounds remained on the cells for the duration of the experiment and at regular intervals the virus titer was determined by TCID_50_ assay (Figure 2A,B).

At both of the MOI tested, there was a >10-fold reduction in final virus titers in the presence of nystatin, as anticipated [33]. However, we also observed a ~10-fold reduction in virus titers in the presence of 25-HC, further suggesting MNV replication is sensitive to this oxysterol.

To confirm the inhibitory effect of 25-HC on virus replication, we employed a fluorescent focus assay to measure the number of cells expressing the viral non-structural proteins after infection in the presence of 25-HC. RAW264.7 cells infected with MNV were treated immediately with nystatin or 25-HC in addition to an ethanol-only control. At 12 h post-infection, the number of cells expressing the viral non-structural protein 1/2 (NS1/2) was quantified by indirect immunofluorescence and enumerated using an IncuCyte Zoom FLR using the integrated software (Figure 2C).

In line with data from previous studies, the addition of nystatin resulted in a significant decrease in the number of cells expressing viral non-structural proteins [34]. In agreement with the viral growth curve (Figure 2A,B) the addition of 25-HC demonstrated a significant decrease in the number of cells expressing viral non-structural protein, which is indicative of a reduction of viral replication.

To determine any effects of 25-HC on the viral capsid, the virus was pre-incubated with increasing concentrations of 25-HC for 4 h at 37 °C before titration on RAW264.7 cells (Appendix A). Pre-treating the virus with 25-HC before titration did not reduce viral titers, suggesting that the observed reduction in replication was not due to direct effects on the viral nucleocapsid.

#### 3.2. Pre-Treatment with 25-HC Prior to Virus Infection Reduced MNV Replication

The above data suggest that MNV replication is inhibited in the presence of 25-HC when added simultaneous to the addition of virus. We then sought to identify the optimal time for the addition of exogenous 25-HC to inhibit viral replication in a time-of-addition study. Replicate wells of RAW264.7 cells were pre-treated with either nystatin or 25-HC (or an ethanol-only control) at regular time intervals before infection with MNV at either 0.05 or 5 TCID_50_/cell. In addition, the cells were infected with MNV and treated with nystatin or 25-HC at 4 h post-infection after the removal of any viruses that had not entered the cells. Subsequently, cells were infected with MNV and immediately treated with 25-HC or nystatin, in a similar way as was undertaken in Figure 2. At 48 h post-infection, the total number of viruses was collected from the cells and supernatant combined and titrated by TCID_50_ assay on RAW264.7 cells (Figure 3).

There was a 10–100-fold significant decrease in virion production at both MOI when the cells were treated with nystatin at 4 h pre-infection, in addition to when the cells were treated simultaneously to infection. This decrease was more prominent when the cells were pre-treated 4 h prior to infection and with the lower MOI used in the experiment. There was no significant decrease in the virus titer when the cells were treated with nystatin 4 h after infection. The same pattern of inhibition was also observed upon the addition of 25-HC; with a ~10-fold, significant, decrease in virus production when the cells were treated 16 or 4 h before infection or with simultaneous treatment (i.e. 0 h). Again, the most pronounced decrease in titers was observed at the lower MOI and after pre-treatment for 4 h before infection. Overall these data are consistent with the data in Figure 1 and Figure 2, suggesting 25-HC can antagonise MNV replication.

#### 3.3. 25-HC Stimulates Caspase Activation and Cell Death in MNV-Infected Cells

Treating RAW264.7 cells with 25-HC alongside-to or prior-to infection with MNV significantly reduced virus replication. MNV replication is known to induce caspase-dependent apoptosis resulting in cell death, which has been postulated to be a mechanism that aids virus release or control NS1/2 cleavage to regulate intestinal persistence [35,36,37,38,39]. In many cell types including macrophages, 25-HC has also been shown to induce apoptosis through multiple cellular pathways resulting in cell death [36,40]. We therefore speculated that the 25-HC-induced reduction in virion production (seen in Figure 1, Figure 2 and Figure 3) was due to increased or accelerated apoptosis in MNV-infected cells, which resulted in cell death at a non-optimal time for virion production. To investigate the combined effects of 25-HC and MNV on caspase 3/7 activity, we carried out a fluorescent caspase 3/7 assay. RAW264.7 cells were infected with MNV in the presence or absence of 25-HC at the same MOI that was used in Figure 3 and cell death was monitored over 24 h using an IncuCyte ZOOM FLR (Figure 4). 

In agreement with previous studies, MNV infection at the higher MOI induced caspase 3/7 activity, which peaked and plateaued between 16 and 20 h post-infection. The addition of 25-HC alone also induced RAW264.7 cell apoptosis, which was less pronounced than the MNV-induced apoptosis at the higher MOI (5 TCID_50_/cell), but more pronounced than that induced at the lower MOI (0.05 TCID_50_/cell). However, in both cases, there appeared to be increased caspase 3/7 activation at earlier time points when both MNV and 25-HC were added in combination. It should also be noted that the caspase levels in the mock cells are higher than anticipated suggesting the experimental setup may not be optimal (for example initial cell seeding density too high). Despite these caveats, these data suggest that 25-HC accelerated the induction of caspase 3/7 in MNV-infected RAW264.7 cells.

We hypothesised that the additive increase in caspase activity when 25-HC and MNV were present together would correlate with more cell death, compared to either virus or to oxysterol alone. To study this, we chose to conduct a MTS assay to measure the number of viable cells after infection with MNV directly in the presence of 25-HC. RAW264.7 cells were therefore treated with a serial dilution of MNV in the continuous presence of 25-HC before assessment of the number of viable cells using the Cell Titer 96 AQueous One reagent. To confirm that any observed effect was specific to 25-HC the experiment was also repeated with 22-S-hydroxycholesterol (22-S-HC), cholesterol or nystatin (Figure 5A). 

Consistent with our hypothesis, we observed a significant 60% decrease in cell viability when RAW264.7 cells were infected with MNV in the presence of 25-HC, compared to with the virus or 25-HC alone. Furthermore, this reduction was specific to 25-HC and neither cholesterol or 22-S-HC demonstrated any reduction or MNV-induced cytotoxicity. There was a small but non-significant decrease in viability with MNV infection in the presence of nystatin, despite the previously observed reduction in virion production with nystatin at this concentration. 

#### 3.4. 25-HC Inhibits HSV-1 Replication in RAW264.7 Macrophages

Taken together, our data suggest that the oxysterol, 25-HC, increased MNV-induced apoptosis while reducing infectious virus production. Previous reports have demonstrated the inhibitory effect of 25-HC on HSV-1 replication in several cell lines (including HeLa [13], HEK293T [12] and Vero cells [15]). Furthermore, RAW264.7 cells should be permissive for HSV-1.

Therefore, we sought to utilise HSV-1 to confirm the response of RAW264.7 to the sterols used in this study. To measure HSV-1 replication we employed a virus expressing GFP (HSV-1–GFP), which can therefore be used as a surrogate marker for viral replication. RAW264.7 cells were infected with HSV-1–GFP in the presence of exogenous 25-HC, 22-S-HC or cholesterol as well as nystatin, a known inhibitor of HSV-1, as a control [41]. Viral replication was monitored by GFP expression and enumerated using an IncuCyte ZOOM Dual Colour FLR (Figure 5B).

Consistent with previous reports, 25-HC and nystatin demonstrated an approximately 60% and 80% inhibitory effect on HSV-1 replication in RAW264.7 cells, respectively. The inhibitory effect of 22-S-HC was unexpected. However, the addition of unmodified cholesterol had no effect on HSV-1 replication, in agreement with previous studies [42,43].

## 4. Discussion

Understanding virus–host interactions can be important for the future control of viral infection. Compounds with broadly acting anti-viral activities offer promising candidates for therapeutic developments. Of interest are compounds that do not target virus replication directly but stimulate innate immune effector molecules that are naturally broadly acting. However, the effect of 25-HC on the replication of noroviruses has not been elucidated. In this study, we have begun to investigate the effect of adding exogenous 25-HC on norovirus replication, focusing on MNV as a safe and robust in vitro model.

In our study, the addition of exogenous 25-HC reduced the replication of MNV in a single-cycle growth experiment, fluorescent focus assay and a time-of-addition assay. In these experiments, the reduction in replication observed was between 2- and 10-fold, depending on the time-of-addition relative to infection and the multiplicity of infection used in the experiments. The greatest inhibitory effects were observed at the lower MOI and upon addition 4 h before infection, whereas there was a more modest reduction in replication when 25-HC was added 4 h post-infection (when the majority of viral endocytosis is complete [33]). These observations suggest that the antagonistic effect of 25-HC on MNV replication could occur at multiple steps of the viral replicative-cycle, as has been reported with other viral pathogens [44]. However, without knowing the time required for oxysterols to cross the cellular membrane, it is difficult to draw firm conclusions as to which stage of the viral replicative-cycle the compound primarily targets.

In our dose–response experiments, 25-HC had an IC_50_ value of ~2.8 µM and CC_50_ of ~50 µM with a selectivity index of ~7. In comparison, the selectivity index for nystatin calculated in these assays was ~3, which is similar to that reported for other viruses [45]. However, for the majority of our experiments we choose to use one concentration of 25-HC (6.6 µM), which is similar to the concentrations used in other studies (1–10 µM) [11,16]. Furthermore, at this concentration, minimal cytotoxicity was observed for RAW264.7 cells when oxysterol was present in isolation. However, both caspase activity assays and MTS assays suggested that the addition of 25-HC simultaneous to MNV infection, stimulated virus-induced apoptosis and reduced the number of viable cells in the cultures (as a result of increased cell death or reduced cell proliferation). In addition, the effect of 25-HC on norovirus replication was not shared by the other sterols assayed here. This was in contrast to HSV-1 replication, which was inhibited in response to both 25-HC as well as 22-S-HC. Previous reports have also demonstrated the inhibitory effect of both 25-HC and 27-HC on the replication of HSV-1 [12,13,15]. Thus, these data suggest that the response of MNV to 25-HC may be oxysterol-specific.

Several groups have documented that MNV infection induces apoptosis by caspase 9 and 3 activation through mitochondrial pathways via a mechanism that is not fully described [32,35,36]. 25-HC has also been demonstrated to induce apoptosis, thought to be through the accumulation of endoplasmic reticulum stress, leading to pro-apoptotic signaling via Jun N-terminal kinase (JNK), transcriptional factor C/EBP homologous protein (CHOP) and the B cell lymphoma gene 2 (Bcl-2) family of proteins [36,40,46]. Thus, it is possible that the combined pro-apoptotic stimuli from both MNV and 25-HC in combination increased apoptosis when compared to either stimulus alone. MNV induces apoptosis which peaks at 16–20 h post infection [32,35,47]. Therefore, it is possible that the premature induction of apoptosis during MNV infection is detrimental to viral replication, thus offering mechanisms behind which 25-HC inhibited virion production here. Many viral infections are known to initiate cellular apoptosis or necroptosis by activating an array of cellular signaling pathways (as reviewed by [48]). Studies, both in vivo and in vitro, have demonstrated that apoptosis plays a critical role in the host defense against infections and suppression of viral replication. Although apoptosis is a poor inducer of host immunity, it is not inconceivable that such an apoptotic mechanism would exist to counteract MNV replication. Further investigation would be required using cell death specific assays together with experiments to dissect cellular pathways to fully elucidate the interaction between MNV and 25-HC. Furthermore, simultaneous measurements of virus production and cell survival at the same timepoint would help clarify the relationship between MNV replication and cell survival.

It is also essential to note that both MNV and 25-HC induce a wide array of changes to cellular biochemistry likely including both pro- and anti-apoptotic stimuli, thus it is important to consider other potential modes-of-action. These could be via alterations to intracellular membranes, the composition of which can be altered by 25-HC production, or via broader type I interferon responses that can upregulate 25-HC production [49]. MNV infection upregulates interferon responses [50] and sensitivity to interferon has been shown in vivo and in vitro [51]. This sensitivity may be conferred in part by the action of 25-HC. Further investigation is therefore required to elucidate the mechanism by which 25-HC inhibits the replication of norovirus.

The concentration of 25-HC used in this study is higher than that found in the blood of healthy individuals, i.e., nanomolar range [52,53]. However, elevated 25-HC levels are noted in the development of several chronic disease (such as liver diseases) and excessive 25-HC is also related to the development of atherosclerosis and other diseases involving macrophage apoptosis [46,54,55]. In future studies, it may also be relevant to assay the effect of the physiological concentrations of 25-HC (both normal and disease-related) on viral replication. In addition, studying the response of HNV model systems, such as the recently developed B-cell or enteroid cultures systems, to oxysterol treatment could be important. If 25-HC demonstrates an inhibitory effect on HNV at physiological concentrations, this could be investigated as a future therapeutic strategy against these viruses. 

## Figures and Tables

**Figure 1 viruses-11-00097-f001:**
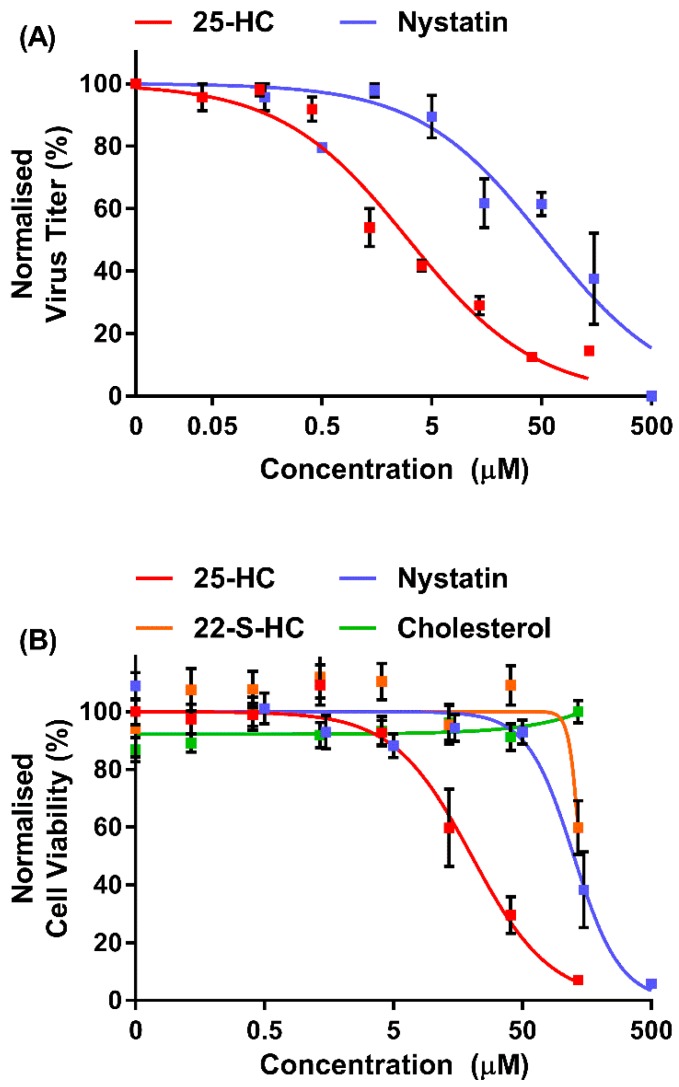
In vitro anti-viral activity of 25-HC on MNV replication. (**A**) RAW264.7 cells were infected with MNV at a multiplicity of infection (MOI) of approximately 0.05 TCID_50_/cell and were immediately treated with a range of concentrations of 25-HC (up to 135 µM) or nystatin (up to 500 µM) or the ethanol-only control. At regular intervals, replicate wells were harvested, the supernatant clarified by centrifugation and the virus titer determined by TCID_50_ assay. Data show mean virus titer (*n* = 2 ± SEM). (**B**) RAW264.7 cells were incubated with the indicated compounds for 72 h before the cell viability was measured by MTS assay. The ethanol (EtOH) or dimethyl sulphoxide (DMSO) solvents were used as controls. Data are expressed as mean percentage cell viability and normalized to untreated cells (*n* = 3 ± SEM).

**Figure 2 viruses-11-00097-f002:**
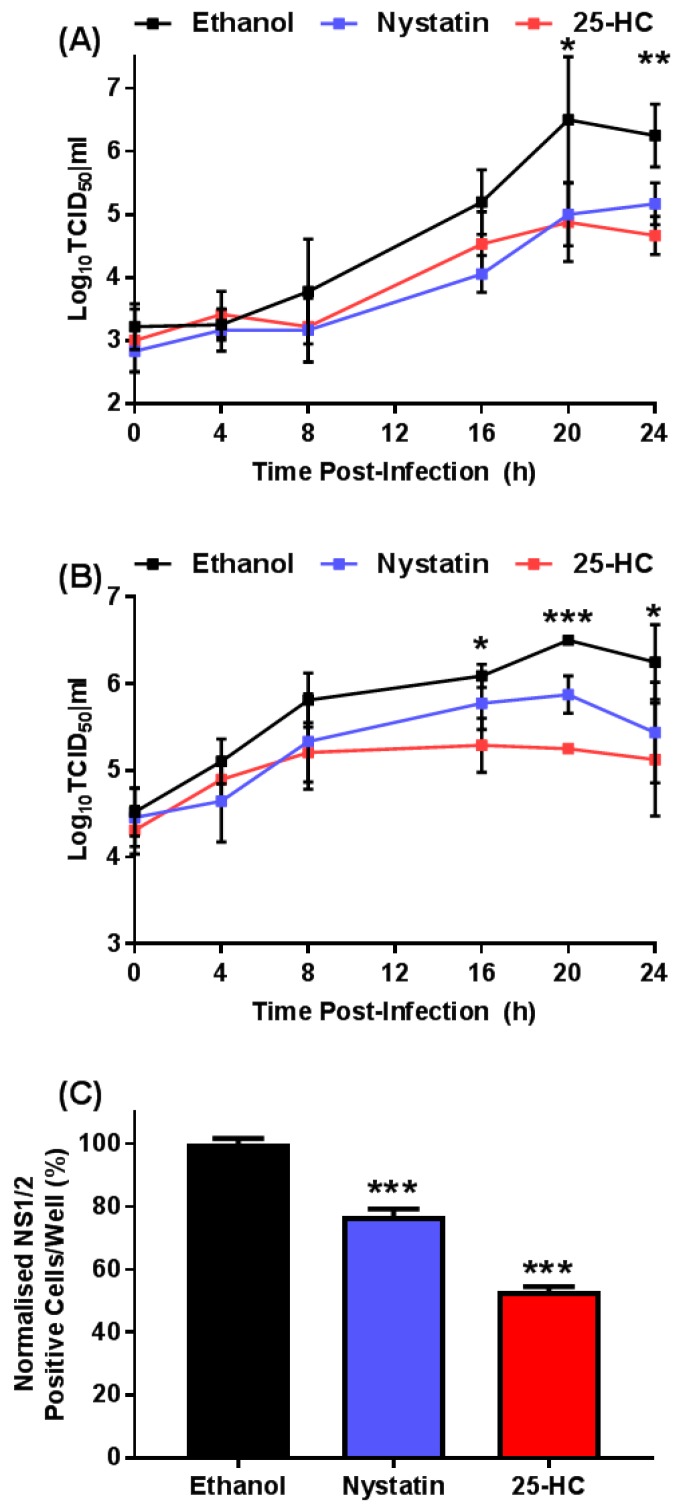
The oxysterol, 25-HC, reduced the titer of MNV. RAW264.7 were infected with MNV at a MOI of approximately (**A**) 0.05 TCID_50_/cell or (**B**) 5 TCID_50_/cell and immediately treated with 25-HC (6.6 µM), nystatin (50 µM) or ethanol-only control. At regular intervals, replicate wells were harvested, supernatant clarified by centrifugation and virus titer determined by TCID_50_ assay. Data show mean virus titer (*n* = 3 ± SEM, * = *p* < 0.05, ** = *p* < 0.01, *** = *p* < 0.001). (**C**) RAW264.7 cells were infected with MNV and immediately treated with 25-HC (6.6 µM), nystatin (50 µM) or ethanol-only control. At 12 h post-infection, the expression of MNV NS1/2 protein was detected by indirect immunofluorescence and the number of virus positive cells was quantified using an IncuCyte ZOOM FLR. Data show the mean number of virus-infected cells per well and statistical analysis compared to ethanol-only control (*n* = 3 ± SEM, *** = *p* < 0.001).

**Figure 3 viruses-11-00097-f003:**
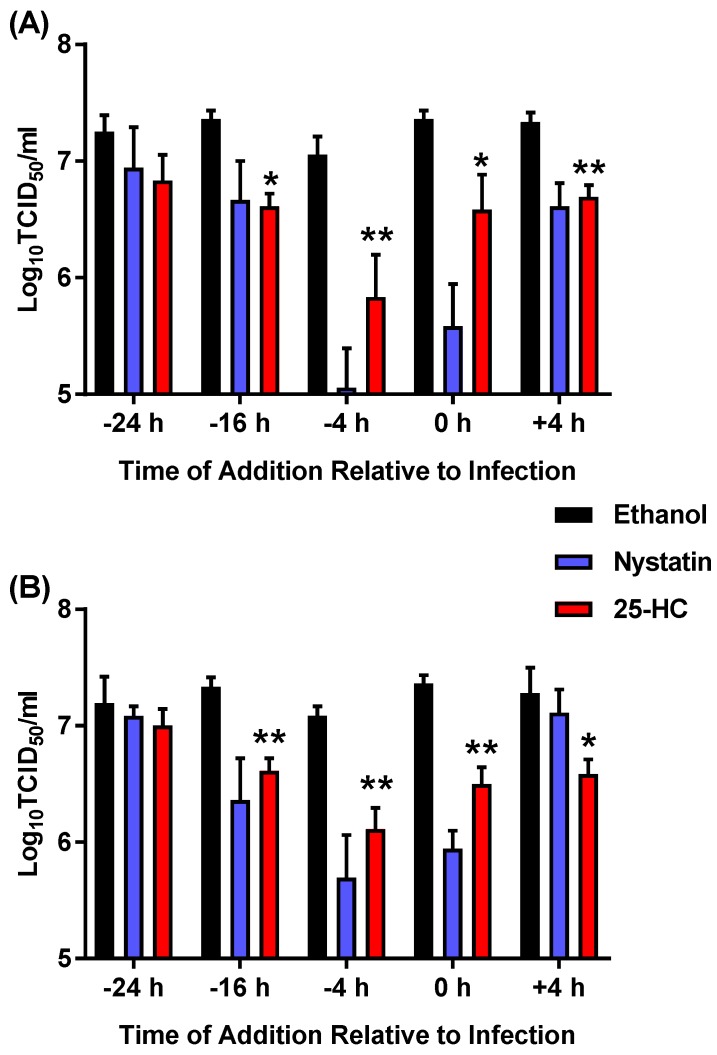
Pre-treating RAW264.7 cells with 25-HC inhibits MNV replication. RAW264.7 cells were pre-treated with 25-HC (6.6 µM), nystatin (50 µM) or the ethanol-only control at 24, 16 or 4 h before infection (−24 h, −16 h or −4 h, respectively), and the compounds were removed prior to the addition of MNV at (**A**) 0.05 TCID_50_/cell or (**B**) 5 TCID_50_/cell. Subsequently, the cells were infected with MNV and immediately treated with the compounds (0 h) or treated at 4 h post-infection (+4 h) after removal of the virus. At 48 h post-infection, the supernatants were removed, clarified by centrifugation and production of infectious virus determined by TCID_50_ assay on RAW264.7 cells. Data show mean virus titer and statistical analysis of 25-HC by comparison to the ethanol-only control (*n* = 3 ± SEM, * = *p* < 0.05, ** = *p* < 0.01).

**Figure 4 viruses-11-00097-f004:**
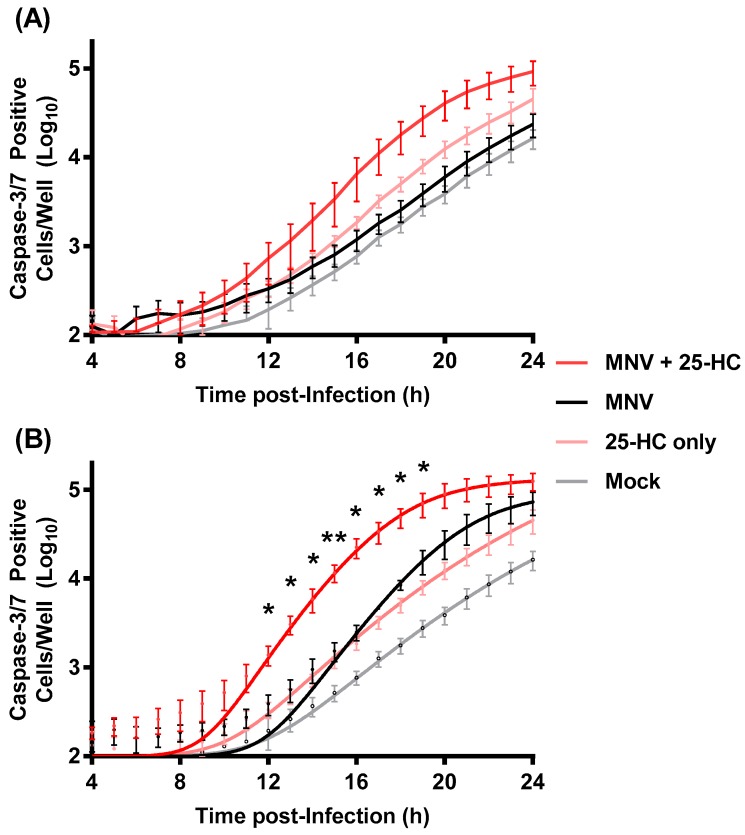
MNV-induced apoptosis in the presence of 25-HC. Replicate wells of RAW264.7 cells were infected with MNV at an MOI of (**A**) 0.05 or (**B**) 5 TCID_50_/cell and immediately treated with 25-HC (6.6 µM). As controls, the replicate wells were infected with MNV without the addition of 25-HC or treated with 25-HC alone. To quantify apoptosis, the caspase 3/7 assay reagent was added directly to the wells immediately after infection and the caspase activity was measured by fluorescence at regular intervals using IncuCyte ZOOM FLR. Data are shown as mean caspase-3/7 positive cells per well and statistical analysis of MNV with 25-HC by comparison to the MNV-only sample (*n* = 4 ± SEM, * = *p* < 0.05, ** = *p* < 0.01).

**Figure 5 viruses-11-00097-f005:**
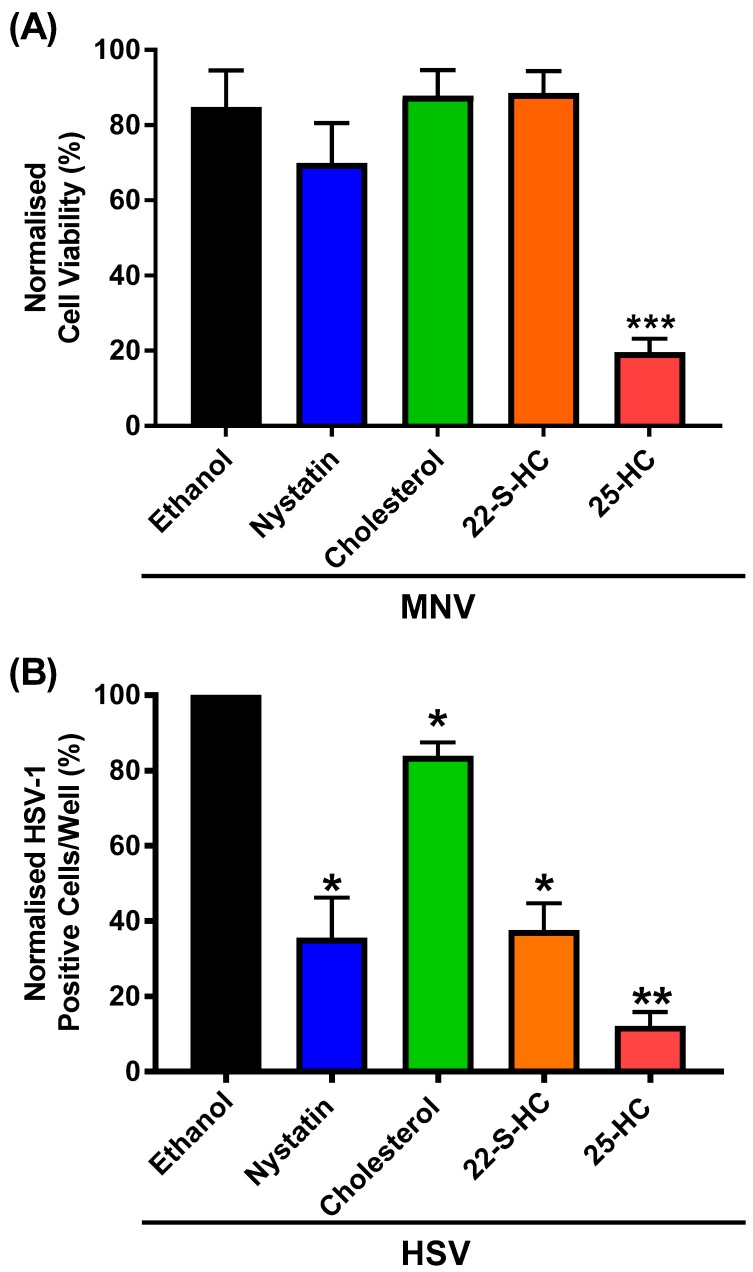
The effect of 25-HC is oxysterol-specific. (**A**) RAW264.7 cells were infected with MNV at 0.05 TCID50/cell in the presence of 25-HC, 22-S-HC, cholesterol (all 6.6 µM), nystatin (50 µM) or ethanol-only control. Cell viability was measured by MTS assay 72 h post-infection. Data are expressed as mean percentage cell viability normalized to untreated and uninfected cells (*n* = 3 ± SEM, *** = *p* < 0.001). (**B**) Replicate wells of RAW264.7 cells were infected with HSV-1–GFP at an MOI of 0.03 PFU/cell and treated with 25-HC, 22-S-HC, cholesterol (all 6.6 µM), nystatin (50 µM) or ethanol-only control. Cells were incubated for 48 h before GFP expression was quantified using an Incucyte ZOOM FLR. Data show the mean number of HSV-1-positive cells per well normalized to ethanol solvent control and statistical analysis compared to ethanol-only control (*n* = 3 ± SEM, * = *p* < 0.05, ** = *p* < 0.01).

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
