# Peer review of "The Oxysterol 25-Hydroxycholesterol Inhibits Replication of Murine Norovirus"

_viruses, 2019, doi:10.3390/v11020097_

Reviewer 1 Report

The manuscript, entitled “The response of Norovirus to the Oxysterol 25-Hydroxycholesterol, describes effects of 25-HC on MNV replication. I believe this manuscript has much room to be improved.

 1.     Major points

1) I think the title has to be changed to reflect the main points of the manuscript. For example, “25-Hydroxycholestrolo inhibits replication of murine norovirus in RAW264.7 cells”

2) The manuscript needs editing by a native English speaker. For example, lines 95-96: “each compounds and virus titration was added” -> virus was added but not virus titration. And each compound was added. Typos and grammatical errors here and there cause confusions.

3) Fig1 A/B: statistical analysis needs to be done.

4) lines 220-222: the authors claimed that 25-HC can antagonize MNV replication based on their observation that 25-HC treatment at -4, 0, 4 h of virus infection significantly inhibited virus titres at 48hr post-infection. However, if early stage of viral infection is affected by 25-HC, then there should be no difference between -4/0 and 4 h of virus infection in a time-of-addition assay. But, there seem to be significant differences between those time points. These data imply that 25-HC may inhibit virus entry rather than viral replication. To discriminate these possibilities, cellular localization of cell culture-added 25-HC, fluorescence-labeled 25-HC may help.

5) Fig 3B. Even mock cells displayed comparable levels of caspase-3/7. Experimental design for this experiment may not be optimal.

6) in Fig4A, effects of 25-HC seem to be promoting, not inhibiting, MNV replication, which is opposite of those data presented in Figs 1 and 2. Thorough discussion is entailed.

2.     Minor points

1)     Fig. 1: FFU by IFA was determined at 12 h post-infection/treatment (Fig1C) although TCID50 was measured at 0, 4, 8, 16, 20, and 24 h. Any particular reasons? (Fig1 A/B)

2)     Line 190: define MTS and use.

Author Response

Response to Reviewers

Reviewer 1

Major points

1) I think the title has to be changed to reflect the main points of the manuscript. For example, “25-Hydroxycholestrolo inhibits replication of murine norovirus in RAW264.7 cells”.

We have changed the title to ‘The Oxysterol 25-Hydroxycholesterol Inhibits Replication of Murine Norovirus’ which we believe accurately reflects the outcome of our study.

2) The manuscript needs editing by a native English speaker. For example, lines 95-96: “each compounds and virus titration was added” -> virus was added but not virus titration. And each compound was added. Typos and grammatical errors here and there cause confusions.

We have carefully checked the manuscript to ensure the editing is in correct English.

3) Fig1 A/B: statistical analysis needs to be done.

We have added statistical analysis to the original Figure 1A and B (now Figure 2A and B in the resubmitted manuscript).

4) Lines 220-222: the authors claimed that 25-HC can antagonize MNV replication based on their observation that 25-HC treatment at -4, 0, 4 h of virus infection significantly inhibited virus titres at 48hr post-infection. However, if early stage of viral infection is affected by 25-HC, then there should be no difference between -4/0 and 4 h of virus infection in a time-of-addition assay. But, there seem to be significant differences between those time points. These data imply that 25-HC may inhibit virus entry rather than viral replication. To discriminate these possibilities, cellular localization of cell culture-added 25-HC, fluorescence-labelled 25-HC may help.

The reviewer is correct to note that there are differences between the inhibitory effect observed by adding exogenous 25-HC at -4/0h pre-infection or 4 hours post infection. We agree with the comment raised by this reviewer and reviewer 3 (comment 5) that it is difficult from the data presented here to firmly conclude the point or points at which the virus replicative-cycle is inhibited. We have modified and expanded the results and discussion sections accordingly (lines 277-278 and 356-366). However, we believe that to determine the precise point at which 25-HC prevents replication of virus is outside the scope of this study.

5) Fig 3B. Even mock cells displayed comparable levels of caspase-3/7. Experimental design for this experiment may not be optimal.

Although we have tried to optimise this experiment, we appreciate that the caspase levels for mock infected cells are higher than anticipated. We have added a comment in the text to this effect.

6) In Fig4A, effects of 25-HC seem to be promoting, not inhibiting, MNV replication, which is opposite of those data presented in Figs 1 and 2. Thorough discussion is entailed.

We apologise for the confusion caused by Figure 4A. In this experiment MNV was titrated directly in the presence of 25-HC, which was incubated with the cells for the duration of experiment. Titrating MNV in the constant presence of 25-HC resulted in “additive” cell death which otherwise mimics CPE and thus appears as an increase in viral replication.

To add support to this hypothesis we have conducted a cell viability (MTS) assay with MNV in the presence of 25-HC. Incubating RAW264.7 cells with a low MOI of MNV (0.05 TCID50/cell) together with sub-cytotoxic concentrations of 25-HC resulted in a significant 60% increased cell death compared to 25-HC or virus alone. This new data has been used to replace the previous figure 4A (in the revised manuscript figure 5A), together with an explanation of the material and methods (paragraph 2.10, lines 167-176) and description of the results (Lines 309-325).

Minor points

1)  Fig. 1: FFU by IFA was determined at 12 h post-infection/treatment (Fig1C) although TCID50 was measured at 0, 4, 8, 16, 20, and 24 h. Any particular reasons? (Fig1 A/B).

This was for pragmatic reasons.

2) Line 190: define MTS and use.

The acronym MTS has been defined at its first use in the revised manuscript (line 93-108) in the material and methods section, and we have provided greater detail of its use (to measure cell viability). This is in combination with replacing the original MTS data (Figure S1 in the original submission), with an extended MTS experiment (now Figure 1B in the revised manuscript), to allow us to estimate CC50 concentrations, in response to Reviewer 3 (comments 1-3).

Reviewer 2 Report

This is a well-written report on the response of norovirus to the oxysterol 25- hydroxycholesterol.

I have several suggestions which may improve the quality of the manuscript.

1. In case of HSV-1, abbreviations should be defined in parentheses the first time they appear in the abstract.

2. Material and Methods

- Is there any reason for selected HSV-1 as control?

3. Discussion :

- Please describe more clearly the significance of the results.

- Please describe previous studies for anti-viral effect against a diverse viruses by 25-HC.

Author Response

Reviewer 2

1. In case of HSV-1, abbreviations should be defined in parentheses the first time they appear in the abstract.

The acronym HSV-1 has been defined after its first use in the abstract and in the main body of the text (line 18).

2. Material and Methods

- Is there any reason for selected HSV-1 as control?

HSV-1 has been used in published studies with 25-HC (e.g. references 12, 13 and 15 in the revised manuscript) and therefore we felt it was an appropriate control to use here.

3. Discussion.

- Please describe more clearly the significance of the results.

The discussion has been modified in several places to include the additional data and we have eluded to the significant of the results here and to the importance of further studies.

- Please describe previous studies for anti-viral effect against a diverse viruses by 25-HC.

More information and references have been added to the introduction and discussion.

Reviewer 3 Report

In this study, the authors target an interesting topic, by investigating “the effect of adding exogenous 25-HC on norovirus replication, focusing on MNV as a safe a robust in vitro model” (lines 296-297). Indeed, while 25HC is acknowledged as an effector of innate immunity against viral infections, its mechanism of action against non-enveloped viral pathogens is not widely investigated.

Nevertheless, several major issues have to be addressed in the present study, including the actual anti-MNV potency of 25HC, its mechanism of action, and its effect on apoptosis.

Major revisions

1.     Lines 83-84: the authors must indicate here which was the “required concentration” of oxysterol used for the viability assays (this is described only in figure S1). Moreover, the untreated controls (100% of viability) must be described: since oxysterols were dissolved in ethanol, untreated controls must have been prepared by incubating cells with culture medium supplemented with equal volumes of ethanol respective to the treated samples. From the figure S1, it appears that the authors have tested several concentrations of 25HC, to maximum of 16.4µM. I suggest to raise further the tested concentrations, in order to find a cytotoxic one, and calculate a CC50 (see below, point 3).

2.     Lines 93-95: in the antiviral assay, the authors have tested one single concentration for each oxysterol (i.e. 6.6µM), which is definitively inadequate to assess the antiviral potency of the molecules. When performing antiviral assays, several dilutions of the investigated molecule must be tested, in order to reach a 100% and a 0% of inhibition, and generate a dose/response curve. Dose-response curves are fundamental to create a causal link between the presence of the compound in the medium and the antiviral effect measured. This is even more important in this case because, taking into account figures 1A, B, and C, it looks like 25HC is not endowed with a remarkable anti-MNV activity. But then again, any conclusion about the actual antiviral potency of 25HC is prevented by the lack of a dose-response curve (6.6µM could be a suboptimal concentration).

3.     Dose-response curves are necessary to calculate fundamental parameters. Indeed, when studying an antiviral, it is mandatory to provide (A) the concentrations of the molecule that inhibit the viral infectivity to 50% and to 90% (EC50, and EC90, respectively), (B) the 50% cytotoxic concentration of the molecule (CC50), and (C) the selectivity index resulting from the ratio CC50/EC50. All these values can be drawn by dose-response curves, by using an appropriate software (e.g. GraphPad PRISM).

4.     Lines 96-97: as indicated for cell viability assays (point 1), the authors must describe how the untreated controls were prepared (from Results and figure 1A and B we can realize that ethanol was used for the untreated samples, but this must be included in the Materials and Methods section).

5.     Lines 101-107: for time of addition assay, the viral progeny is harvested 48 hours post infection, when multiple cycles of viral replication took place, making it hard to draw any conclusion about the putative step of viral replication inhibited by 25HC. To perform a time of addition assay, a fluorescent focus reduction assay (that takes into account a single replication cycle) would be more proper. Moreover, the significant (yet not remarkable) inhibition we see (figure 2) when treatment is performed 4 hours before infection, is not sufficient to conclude that “25-HC can antagonise MNV replication, potentially at an early stage of the viral life-cycle” (lines 221-222). Indeed, given the ability of oxysterols to cross cellular membrane and get access to the cytosol, we can speculate that they can modify some intracellular pathway or structure that inhibits later steps of viral replication, rather than the early ones (25HC is endowed with this mechanism against some picornaviridae, for example).

6.     When studying the putative target of an antiviral molecule, it is important to investigate its direct effect on the viral particle, by a virus inactivation assay. This experiment is not included, so we cannot rule out the possibility that 25HC targets MNV nucleocapsid, rather than a cellular determinant.

7.     Line 251: I think that the use of the term “synergistic” is improper. The results rather suggest an additive effect. Moreover, a synergism should always be demonstrated, for example with an isobologram.

8.     Lines 251-258: It is hard to understand the rationale of this experiment. The authors want to investigate if “the increase in caspase activity when 25-HC and MNV were present together would correlate with an increase in cell death”, by performing a TCID50 assay. However, the measurement of TCID50/ml is useful to monitor the production of viral progeny, but gives no information about the rate of cell death. Moreover, the results presented in figure 4A (line 259) are difficult to assemble with the previous ones: it looks like 25HC improves the production of viral progeny, rather than inhibiting it. If this is correct, the authors must evaluate the CPE on 25HC-treated/infected cells, because these results remarkably confute the demonstration that 25HC has a protective effect, and invalidate the premise of a protective effect of 25HC.

9.     It is not clear what the authors intend for “Apparent” when referring to Log10TCID50/ml (Figure 4A, y axis).

Minor revisions

1.     In the Introduction section, when the authors describe the step of non-enveloped virus’ replication inhibited by 25HC (lines 53-56), the following study should be also quoted:

Civra A, Francese R, Gamba P, Testa G, Cagno V, Poli G, Lembo D. 25-Hydroxycholesterol and 27-hydroxycholesterol inhibit human rotavirus infection by sequestering viral particles into late endosomes. Redox Biol. 2018 Oct;19:318-330. doi: 10.1016/j.redox.2018.09.003.

In this paper, the authors disclose the mechanism of action of both 25HC and 27-hydroxycholesterol (27HC) against human rotavirus (a non-enveloped virus).

2.     The expression “viral life-cycle” (widely used all across the paper) should be replaced with “viral replicative cycle”, since viruses are not living beings.

3.     Line 78, 99, 172, 219, 272: The word “Titre” should be replaced with “Titer”.

4.     Line 126: “NS/12” must be replaced with “NS1/2”

5.     Line 108: I suggest to shift this experiment (“Single cycle growth analysis”) before paragraph 2.4. I think it would be more logic; actually, this experiment is the first one described in the results section (line 151).

6.     Line 111: “treated” must be replaced with “infected”

7.     Line 162: In figure 1A, a SEM upper bar is outside the axis limit

8.     Line 111: the authors must indicate which is “the indicated MOI” used (this information is present only in the Results section). Moreover, the acronym MOI is used here for the first time, so it must be clarified.

9.     Lines 155-161: the informations about methods included here are already quoted in part in paragraph 2.6. I suggest to shift the necessary informations about methods from paragraph 3.1 to paragraph 2.6, and eliminate the redundant ones.

10.  Lines 175-183: as above, the informations about methods included here are already quoted in part in paragraph 2.7. I suggest to shift the necessary informations about methods from paragraph 3.1 to paragraph 2.7, and eliminate the redundant ones.

11.  Lines 195-202: as above, the informations about methods included here are already quoted in part in paragraph 2.5. I suggest to shift the necessary informations about methods from paragraph 3.2 to paragraph 2.5, and eliminate the redundant ones.

Author Response

Reviewer 3

Major revisions

1. Lines 83-84: the authors must indicate here which was the “required concentration” of oxysterol used for the viability assays (this is described only in figure S1). Moreover, the untreated controls (100% of viability) must be described: since oxysterols were dissolved in ethanol, untreated controls must have been prepared by incubating cells with culture medium supplemented with equal volumes of ethanol respective to the treated samples. From the figure S1, it appears that the authors have tested several concentrations of 25HC, to maximum of 16.4µM. I suggest to raise further the tested concentrations, in order to find a cytotoxic one, and calculate a CC50 (see below, point 3).

2. Lines 93-95: in the antiviral assay, the authors have tested one single concentration for each oxysterol (i.e. 6.6µM), which is definitively inadequate to assess the antiviral potency of the molecules. When performing antiviral assays, several dilutions of the investigated molecule must be tested, in order to reach a 100% and a 0% of inhibition, and generate a dose/response curve. Dose-response curves are fundamental to create a causal link between the presence of the compound in the medium and the antiviral effect measured. This is even more important in this case because, taking into account figures 1A, B, and C, it looks like 25HC is not endowed with a remarkable anti-MNV activity. But then again, any conclusion about the actual antiviral potency of 25HC is prevented by the lack of a dose-response curve (6.6µM could be a suboptimal concentration).

3. Dose-response curves are necessary to calculate fundamental parameters. Indeed, when studying an antiviral, it is mandatory to provide (A) the concentrations of the molecule that inhibit the viral infectivity to 50% and to 90% (EC50, and EC90, respectively), (B) the 50% cytotoxic concentration of the molecule (CC50), and (C) the selectivity index resulting from the ratio CC50/EC50. All these values can be drawn by dose-response curves, by using an appropriate software (e.g. GraphPad PRISM).

Response to comments 1-3:

The aim of our study was to identify whether MNV replication was sensitive to exogenous 25-HC. Therefore our initial experimental approach was to simply test MNV for sensitivity against the maximum concentration of 25-HC that was tolerated by RAW264.7 cells before any significant cytotoxicity was observed (i.e. 6.6µM). However we agree that from this data alone, it is not possible to calculate the potential anti-viral potency of this molecule. Thus, we have now conducted antiviral and cell viability assays using a range of 25-HC concentrations and use this information to calculate EC50 and CC50 values as suggested by the reviewer. Our new data suggest that the selectivity index of 25-HC is greater than that of nystatin, a known MNV inhibitor. Once again, the maximum concentration tested that was tolerated before significant cytotoxicity is observed was 6.6 µM. Therefore, we believe that this is an appropriate concentration to use for the more detailed studies later in the manuscript. Details of the conditions for the solvent-only control samples have also been included in the materials and methods sections, as appropriate.

This data has been included as a new figure 1 within the revised manuscript (and replaces the original Figure S1). A description of the method has been included in the materials and methods section and description/discussion in the results section.

4. Lines 96-97: as indicated for cell viability assays (point 1), the authors must describe how the untreated controls were prepared (from Results and figure 1A and B we can realize that ethanol was used for the untreated samples, but this must be included in the Materials and Methods section).

Further information has been added into each paragraph of the materials and methods sections to clarify these points.

5. Lines 101-107: for time of addition assay, the viral progeny is harvested 48 hours post infection, when multiple cycles of viral replication took place, making it hard to draw any conclusion about the putative step of viral replication inhibited by 25HC. To perform a time of addition assay, a fluorescent focus reduction assay (that takes into account a single replication cycle) would be more proper. Moreover, the significant (yet not remarkable) inhibition we see (figure 2) when treatment is performed 4 hours before infection, is not sufficient to conclude that “25-HC can antagonise MNV replication, potentially at an early stage of the viral life-cycle” (lines 221-222). Indeed, given the ability of oxysterols to cross cellular membrane and get access to the cytosol, we can speculate that they can modify some intracellular pathway or structure that inhibits later steps of viral replication, rather than the early ones (25HC is endowed with this mechanism against some picornaviridae, for example).

As noted by reviewer 1 (comment 4) it is difficult from the data presented here to firmly conclude the point or points at which the virus replicative-cycle is inhibited. Furthermore, we believe that precise determination of the step at which 25-HC inhibits replication of MNV is outside the scope of this study. We agree with the points raised by the reviewer and have modified and expanded the results and discussion sections according (lines 269-278 and 356-366).

6. When studying the putative target of an antiviral molecule, it is important to investigate its direct effect on the viral particle, by a virus inactivation assay. This experiment is not included, so we cannot rule out the possibility that 25HC targets MNV nucleocapsid, rather than a cellular determinant.

The reviewer is correct to point out that in the original submission we did not investigate the direct effect of 25-HC on the viral particle. In response we have investigated the effect of pre-incubating un-purified stocks of MNV with increasing concentration of 25-HC for 4 hours at 37°C before titration by TCID50 on RAW264.7 cells. Pre-incubation of MNV with concentration of 25-HC up to 135 µM (approximately 20-fold higher concentrations than used in the majority of our experiments) had no effect on virus titres, suggesting 25-HC does not target the viral nucelocapsid.

This data has been included as a supplementary figure within the revised manuscript (Figure S1). A description of the method has been included in the materials and methods section (lines 85-90) and of the data in the results section (lines 234-237).

7. Line 251: I think that the use of the term “synergistic” is improper. The results rather suggest an additive effect. Moreover, a synergism should always be demonstrated, for example with an isobologram.

We agree with this point raised by the reviewer and the use of “synergistic” has been replaced with “additive” in the modified manuscript.

8. Lines 251-258: It is hard to understand the rationale of this experiment. The authors want to investigate if “the increase in caspase activity when 25-HC and MNV were present together would correlate with an increase in cell death”, by performing a TCID50 assay. However, the measurement of TCID50/ml is useful to monitor the production of viral progeny, but gives no information about the rate of cell death. Moreover, the results presented in figure 4A (line 259) are difficult to assemble with the previous ones: it looks like 25HC improves the production of viral progeny, rather than inhibiting it. If this is correct, the authors must evaluate the CPE on 25HC-treated/infected cells, because these results remarkably confute the demonstration that 25HC has a protective effect, and invalidate the premise of a protective effect of 25HC.

9. It is not clear what the authors intend for “Apparent” when referring to Log10TCID50/ml (Figure 4A, y axis).

Reponses to 8 and 9

We apologise for the confusion caused by our axis labelling in Figure 4A. In this experiment MNV was titrated directly in the presence of 25-HC, which was left on the cells for the duration of experiment. Titrating MNV in the constant presence of 25-HC results in an “additive” cell death which otherwise mimics CPE and thus appears as an increase in viral replication, which we describe as an “apparent” increase in TCID50 (to distinguish it from an “real” increase in virus replication).

To clarify this difference (as in response to reviewer 1, point 6), and better demonstrate our hypothesis we have conducted a cell viability (MTS) assay with MNV in the presence of 25-HC. As explained in response to reviewer 1 (point 6) incubating RAW264.7 cells with a low MOI of MNV (0.05 TCID50/cell) together with sub-cytotoxic concentrations of 25-HC results in a significant 60% increased cell death compared to 25-HC or virus alone.

To avoid confusion this new data has been used to replace the previous figure 4A (in the revised manuscript figure 5A), together with an explanation of the material and methods (paragraph 2.10, lines 167-176) and description of the results (Lines 309-325).

Minor revisions

1. In the Introduction section, when the authors describe the step of non-enveloped virus’ replication inhibited by 25HC (lines 53-56), the following study should be also quoted:

Civra A, Francese R, Gamba P, Testa G, Cagno V, Poli G, Lembo D. 25-Hydroxycholesterol and 27-hydroxycholesterol inhibit human rotavirus infection by sequestering viral particles into late endosomes. Redox Biol. 2018 Oct;19:318-330. doi: 10.1016/j.redox.2018.09.003.

In this paper, the authors disclose the mechanism of action of both 25HC and 27-hydroxycholesterol (27HC) against human rotavirus (a non-enveloped virus).

This reference has been included at this section of the introduction (lines 51-52).

2. The expression “viral life-cycle” (widely used all across the paper) should be replaced with “viral replicative cycle”, since viruses are not living beings.

The phrase “viral life-cycle” has been changed for “viral replicative-cycle” throughout the manuscript.

3. Line 78, 99, 172, 219, 272: The word “Titre” should be replaced with “Titer”.

The spelling of this word has been changed.

4. Line 126: “NS/12” must be replaced with “NS1/2”.

This was a typographical error which has been corrected.

5. Line 108: I suggest to shift this experiment (“Single cycle growth analysis”) before paragraph 2.4. I think it would be more logic; actually, this experiment is the first one described in the results section (line 151).

The order of paragraphs 2.1 to 2.10 has been changed to reflect the order of the experiments described in the results section. 

6. Line 111: “treated” must be replaced with “infected”.

This has been changed in the modified manuscript.

7. Line 162: In figure 1A, a SEM upper bar is outside the axis limit.

The axis on the figure has been modified.

8. Line 111: the authors must indicate which is “the indicated MOI” used (this information is present only in the Results section). Moreover, the acronym MOI is used here for the first time, so it must be clarified.

The acronym MOI has been defined after its first use here and details of the multiplicity of infection used in each experiment are now included in the methods section.

9. Lines 155-161: the informations about methods included here are already quoted in part in paragraph 2.6. I suggest to shift the necessary informations about methods from paragraph 3.1 to paragraph 2.6, and eliminate the redundant ones.

10. Lines 175-183: as above, the informations about methods included here are already quoted in part in paragraph 2.7. I suggest to shift the necessary informations about methods from paragraph 3.1 to paragraph 2.7, and eliminate the redundant ones.

11. Lines 195-202: as above, the informations about methods included here are already quoted in part in paragraph 2.5. I suggest to shift the necessary informations about methods from paragraph 3.2 to paragraph 2.5, and eliminate the redundant ones.

Reponses to 9, 10 and 11

The details in the methods section has been shortened to eliminate redundant information.

Round  2

Reviewer 1 Report

the manuscript has been much improved after revision.

One minor thing: Figure1A Y-axis title: Titre -> Titer (best to be consistent throughout the manuscript)

Author Response

Review 1

One minor thing: Figure1A Y-axis title: Titre -> Titer (best to be consistent throughout the manuscript)

The axis on the Figure has been changed.

Reviewer 3 Report

The authors performed supplemental experiments in order to investigate the antiviral activity of 25-hydroxycholesterol (25-HC) against murine norovirus (MNV) on RAW264.7 cells. On one side, these experiments explore the effect of 25-HC on RAW264.7 viability, mostly in terms of cytotoxic concentrations, but on the other side they raise further issues that weaken the significance of the study.

1.           Figure 4, lines 294-300: the results of these experiments are highly arguable, since the authors measure an increase of caspases also in the mock controls, without giving a reasonable explanation about it. This calls into question the whole experiment, which should have been optimized, especially in the light of its central position in demonstrating the authors’ hypotheses.

2.           Figure 5A, lines 319-329: the results of this experiment and its rationale are highly questionable for several reasons.

    -      Firstly, an MTS assay is useful to measure the amount of viable cells in a specific sample (it is actually a “proliferation assay”), so it is quite improper to use it to determine the rate of cell death; the measurment of lactate dehydrogenase test released in the supernatant by dead cells would have been more proper.

    -      The results should have been coupled with the MNV titers (TCID50/ml) produced at 72 hours post infection (i.e. when this experiment was stopped). Since these data are missing, we can make no link between viral replication and cell viability.

    -      For this assay the authors chose to test a concentration of 25-HC corresponding to 6.6 µM and stopped the experiment 72 hours after infection/treatment. Firstly, this concentration is suboptimal since it does not inhibit viral replication to 100%: according to figure 1A (lines 193-201), there is a ~70% inhibition of virion production at this concentration at 48 hours post infection; according to figure 2A (lines 220-230), 25-HC treatment reduces viral titer of ~10 fold at 24 hours post infection, but the amount of MNV produced is yet remarkable (about 5x104 TCID50/ml) so it is likely to determine a relevant cytopathic effect. Moreover this concentration (6.6 µM) is detrimental for RAW 264.7 viability after 72 hours from treatment (about 30% of reduction in cell viability according to figure 1B (lines 193-201)). Taking together these informations, it is not surprising that the authors measure only a 20% of cell viability in the 25-HC treated/infected wells (red bar in figure 5A, line 319): this percentage clearly results from the combined toxicity of 25-HC at 6.6µM and cell death due to MNV replication.

    -      One critical control of the experiment is ambiguous. Indeed, it is puzzling that in the untreated/infected control (black bar in figure 5A, line 319) the authors measured a ~80% of cell viability. Actually, according to figure 2A (line 220), RAW 264.7 infected with 0.05 TCID50/cell (the same inoculum of experiment in figure 5A) produce a viral yield of about 106 TCID50/ml after 24 hours (which is likely to be even higher after 72 hours when experiments in figure 5A were stopped) that should have killed all cells in the well. For this reason it is definitively not likely that after 72 hours from inoculum there is still a 80% of viable RAW 264.7 cells in the untreated/infected wells.

    -      Line 320: the title of figure 5 (“Oxysterol specific effects on MNV replication”) is misleading, since the experiment provides no information about MNV replication.

    -      An untreated control for Nystatin (i.e. with DMSO alone) is apparently lacking.

3.           Figure 1, lines 193-201: the results of the two graphs are difficult to compare, since the first one (i.e. the viral yield reduction assay, figure 1A) is performed 48 hours post infection/treatment while the second one (cell viability assay, figure 1B) is performed 72 hours post treatment.

4.           Figures 2A-B-C (lines 220-230) and 3A-B (lines 261-269): the untreated control for Nystatin (i.e. with DMSO alone) is apparently lacking.

5.           No further experiment is performed in order to provide hints about the putative viral replication step inhibited by 25-HC (a time of addition assay performed 12 hours after inoculum would have helped, but this experiment was not performed).

In the “Discussion” paragraph (lines 393-395) the authors hypothesize that: “It is possible therefore that premature induction of apoptosis during MNV infection is detrimental to viral replication, thus offering mechanisms behind which 25-HC inhibited virion production here”. However, no specific marker of apoptosis activation has been studied (the caspase activation experiment should have been optimized and anyway is definitively non sufficient). By contrast, according to the data provided, the inhibition of the production of viral progeny seems to result from a non-specific toxic effect exterted by 25-HC on RAW 264.7 cells. In other words, the overall message that stems from the results presented is that 25-HC partially inhibits MNV replication because it trivially kills the host cells that MNV exploits to replicate, which is definitively not acceptable as an antiviral mechanism of action.

Author Response

Review 3

Comments and Suggestions for Authors

The authors performed supplemental experiments in order to investigate the antiviral activity of 25-hydroxycholesterol (25-HC) against murine norovirus (MNV) on RAW264.7 cells. On one side, these experiments explore the effect of 25-HC on RAW264.7 viability, mostly in terms of cytotoxic concentrations, but on the other side they raise further issues that weaken the significance of the study.

1. Figure 4, lines 294-300: the results of these experiments are highly arguable, since the authors measure an increase of caspases also in the mock controls, without giving a reasonable explanation about it. This calls into question the whole experiment, which should have been optimized, especially in the light of its central position in demonstrating the authors’ hypotheses.

Additional text has been added into the results and discussion sections to note the limitations with these experiments and how they could be optimised for investigation in the future, as suggested by the editors.

2. Figure 5A, lines 319-329: the results of this experiment and its rationale are highly questionable for several reasons.

-Firstly, an MTS assay is useful to measure the amount of viable cells in a specific sample (it is actually a “proliferation assay”), so it is quite improper to use it to determine the rate of cell death; the measurment of lactate dehydrogenase test released in the supernatant by dead cells would have been more proper.

-The results should have been coupled with the MNV titers (TCID50/ml) produced at 72 hours post infection (i.e. when this experiment was stopped). Since these data are missing, we can make no link between viral replication and cell viability.

-For this assay the authors chose to test a concentration of 25-HC corresponding to 6.6 µM and stopped the experiment 72 hours after infection/treatment. Firstly, this concentration is suboptimal since it does not inhibit viral replication to 100%: according to figure 1A (lines 193-201), there is a ~70% inhibition of virion production at this concentration at 48 hours post infection; according to figure 2A (lines 220-230), 25-HC treatment reduces viral titer of ~10 fold at 24 hours post infection, but the amount of MNV produced is yet remarkable (about 5x104 TCID50/ml) so it is likely to determine a relevant cytopathic effect. Moreover this concentration (6.6 µM) is detrimental for RAW 264.7 viability after 72 hours from treatment (about 30% of reduction in cell viability according to figure 1B (lines 193-201)). Taking together these informations, it is not surprising that the authors measure only a 20% of cell viability in the 25-HC treated/infected wells (red bar in figure 5A, line 319): this percentage clearly results from the combined toxicity of 25-HC at 6.6µM and cell death due to MNV replication.

-One critical control of the experiment is ambiguous. Indeed, it is puzzling that in the untreated/infected control (black bar in figure 5A, line 319) the authors measured a ~80% of cell viability. Actually, according to figure 2A (line 220), RAW 264.7 infected with 0.05 TCID50/cell (the same inoculum of experiment in figure 5A) produce a viral yield of about 106TCID50/ml after 24 hours (which is likely to be even higher after 72 hours when experiments in figure 5A were stopped) that should have killed all cells in the well. For this reason it is definitively not likely that after 72 hours from inoculum there is still a 80% of viable RAW 264.7 cells in the untreated/infected wells.

The use of MTS assays to measure cell survival is a standard approached used in the literature. We have, however, clarified this statement in the text as suggested by the editors.

-Line 320: the title of figure 5 (“Oxysterol specific effects on MNV replication”) is misleading, since the experiment provides no information about MNV replication.

This title has been changed.

-An untreated control for Nystatin (i.e. with DMSO alone) is apparently lacking.

The purpose of this study was to investigate the action of 25-HC, therefore ethanol was the most appropriate control to use as the control. Nystatin is a well described inhibitor of MNV which was solely used as a control. Therefore we believe the solvent control for nystatin (i.e. DMSO) is not required in this situation.

3. Figure 1, lines 193-201: the results of the two graphs are difficult to compare, since the first one (i.e. the viral yield reduction assay, figure 1A) is performed 48 hours post infection/treatment while the second one (cell viability assay, figure 1B) is performed 72 hours post treatment.

Details of the experimental conditions are made clear in the text.

4. Figures 2A-B-C (lines 220-230) and 3A-B (lines 261-269): the untreated control for Nystatin (i.e. with DMSO alone) is apparently lacking.

Please see response to comment 2.

5. No further experiment is performed in order to provide hints about the putative viral replication step inhibited by 25-HC (a time of addition assay performed 12 hours after inoculum would have helped, but this experiment was not performed).

The purpose of this study was to establish if MNV replication was sensitive to 25-HC, as has been shown for other positive-sense RNA viruses. We believe determining the putative step of viral replication or the mechanism of inhibition is outside the remit of this study.

6. In the “Discussion” paragraph (lines 393-395) the authors hypothesize that: “It is possible therefore that premature induction of apoptosis during MNV infection is detrimental to viral replication, thus offering mechanisms behind which 25-HC inhibited virion production here”. However, no specific marker of apoptosis activation has been studied (the caspase activation experiment should have been optimized and anyway is definitively non sufficient). By contrast, according to the data provided, the inhibition of the production of viral progeny seems to result from a non-specific toxic effect exterted by 25-HC on RAW 264.7 cells. In other words, the overall message that stems from the results presented is that 25-HC partially inhibits MNV replication because it trivially kills the host cells that MNV exploits to replicate, which is definitively not acceptable as an antiviral mechanism of action.

Additional discussion and references have been added to the main body of the text regarding cell death as a broad antiviral host defence mechanism that has been shown to act against a wide array of viruses.